# OpenReview forum: "Test-Time Preference Optimization: On-the-Fly Alignment via Iterative Textual Feedback"
_ICML.cc/2025/Conference — ICML 2025 poster_

### Official Review · Reviewer_tpEb · 2025-03-06

**Overall Recommendation:** 4

**Summary:**

This paper introduces a method called **Test-Time Preference Optimization (TPO)**, which aims to improve the alignment of large language models (LLMs) with human preferences during inference. TPO builds upon the previous **TextGrad** framework, extending its capabilities to optimize LLM outputs at test time without the need for retraining. Instead of relying solely on numerical feedback, TPO translates reward signals into textual critiques, which are then used to iteratively refine the model’s responses. The results demonstrate that TPO significantly improves the alignment of both unaligned and aligned models, achieving better performance on a range of tasks, including instruction following, safety, preference alignment, and mathematics. The approach is shown to be efficient, scalable, and capable of adapting to evolving human preferences on the fly.

**Claims And Evidence:**

The claims made in the paper are generally supported by clear and convincing evidence, particularly through empirical results and experiments.

- Claim of improved alignment with human preferences on the fly: The paper demonstrates that the Test-Time Preference Optimization (TPO) method improves the alignment of LLM outputs with human preferences during inference, with empirical evidence showing substantial gains in performance across various benchmarks. The results show that TPO enables unaligned models to surpass their pre-aligned counterparts, providing solid evidence for this claim.

- Claim of scalability and efficiency: The paper argues that TPO scales efficiently with both search width and depth during inference. This claim is supported by experiments that show how the method adapts well to different search configurations and outperforms traditional methods in terms of computational efficiency. Additionally, the authors demonstrate that TPO can significantly improve performance in just a few optimization steps, even in resource-constrained settings.

Overall, the claims are supported by clear empirical results and experiments.

**Essential References Not Discussed:**

No.

**Experimental Designs Or Analyses:**

The experimental designs and analyses presented in the paper appear to be sound and valid. The authors employ a series of reward models, as well as both aligned and unaligned models, to evaluate the effectiveness of the proposed method. These models are tested across a wide range of benchmark datasets, covering various aspects such as instruction following, preference alignment, safety, and mathematical ability. The combination of these diverse evaluation metrics provides strong evidence for the method's effectiveness and robustness.

**Methods And Evaluation Criteria:**

The proposed methods and evaluation criteria make sense for the problem and application at hand.

- **Proposed Method (TPO)**: The method of **Test-Time Preference Optimization (TPO)** is well-suited for the problem of aligning LLM outputs with human preferences during inference without requiring retraining. By utilizing textual feedback and reward models to iteratively refine outputs, TPO addresses the challenge of improving alignment in a resource-efficient manner. The use of textual critiques as a feedback mechanism is a novel and effective approach, as it leverages the innate capabilities of LLMs to process and act on natural language feedback, making it both interpretable and adaptable to different tasks.

- **Evaluation Criteria (Benchmark Datasets)**: The selection of benchmark datasets is appropriate for evaluating TPO’s performance. The paper evaluates TPO across a diverse set of benchmarks that cover various aspects of LLM performance, such as instruction following, preference alignment, safety, and mathematical ability. These benchmarks provide a comprehensive assessment of the model’s ability to align with human preferences, which is central to the method’s goal. Additionally, the comparison with models that have undergone traditional training-time preference optimization (like **DPO** and **RLHF**) further strengthens the evaluation, showing that TPO can achieve similar or better results without retraining.

Overall, the methods and evaluation criteria are well-designed and relevant to the problem of aligning LLM outputs with human preferences during inference.

**Other Comments Or Suggestions:**

The content in the appendix largely repeats some of the experimental settings already presented in the main body. It may be helpful to consider simplifying this section.

**Other Strengths And Weaknesses:**

The paper is very well-rounded and presents its contributions in a clear and structured manner. One of its strengths is the use of well-designed diagrams that effectively illustrate the methodology and experimental results, making complex concepts more accessible. The approach is original in its creative extension of the TextGrad framework and its practical application for achieving human alignment on the fly during inference. The clarity of the writing and the thoroughness of the experiments further enhance the paper's impact. Overall, the paper provides a valuable and innovative contribution to the field.

**Questions For Authors:**

1. Besides the comparison with training-time preference optimization in terms of FLOPs, I would like to understand the real-time effects better. Specifically, how much additional time does a user need to wait to receive results when using this method compared to regular use? Additionally, what is the cost in terms of query token consumption for the extra alignment steps?

2. Is there a way to trigger the alignment process only when it is truly needed, rather than applying it continuously?

3. I would also appreciate a discussion on the limitations of this method. In what scenarios will TPO fail, where training-time methods might still work, and why? It would be helpful to see some bad case analyses to better understand the boundaries and potential pitfalls of this approach.

These questions would help clarify some practical aspects of the method's application and performance, which would influence my evaluation of the paper's effectiveness and real-world usability.

**Relation To Broader Scientific Literature:**

The key contributions of this paper are a direct extension of the previous work **TextGrad**. It further refines and demonstrates the effectiveness of TextGrad in achieving human alignment on the fly. By building on the foundational ideas of using textual feedback for optimization, this paper takes a significant step forward in proving that the proposed **Test-Time Preference Optimization (TPO)** method can improve alignment with human preferences during inference, without the need for retraining the model. This advancement aligns with and extends the broader literature on optimizing AI systems with human feedback, particularly in the context of large language models (LLMs).

**Theoretical Claims:**

The paper does not contain any formal proof.

---

> ### Author Rebuttal · Authors · 2025-04-01
>
> Thank your for your thoughtful and constructive feedback. We are encouraged by the insightful comments and suggestions provided. Regarding your specific concerns:
>
> 1. **Simplifying Appendix to Reduce Overlap with the Main Sections**
>
>     We will simplify and restructure the appendix, clearly delineating between content that appears in the main text and supplementary details.
>
> 2. **Real-time Inference Latency of TPO**
>
>     To evaluate real-world latency, we simulated a production environment by deploying an OpenAI-style completion service using vLLM on 4 A100 GPUs. We hosted both Llama-3.1-70B-SFT and Llama-3.1-70B-DPO, and tested them on the HH-RLHF dataset under full server load to emulate peak usage conditions.
>
>     The average inference time per query and associated compute cost (FLOPs) are summarized below:
>
>     | Model | Training FLOPs (PFLOPs) | Inference FLOPs (PFLOPs) | Avg. Inference Time (s/query) |
>     | --- | --- | --- | --- |
>     | TPO-D2-N5 | 0 | 9.3 | 617/500=1.2 |
>     | BoN-60 | 0 | 16.9 | 1,384/500=2.8 |
>     | DPO | 72840.0 | 0.3 | 95/500=0.2 |
>
>     While TPO requires approximately 5× more inference time per query than a fully trained DPO model, it is still significantly faster than BoN-60. The inference FLOPs for TPO (9.3) compared to DPO (0.3) can reflect the added cost as query token consumption.
>
> 3. **Triggering Alignment only when Necessary**
>
>     One possible solution can be leveraging consistency-based metrics to dynamically determine the required number of TPO iterations.
>
>     For example, in Figure 4, we observe that as TPO steps increase, reward scores become both higher and more stable (i.e., lower standard deviation). This suggests that sharper reward distributions may signal greater model confidence for a given query, indicating the model’s stronger capability on this query. Building on this, the standard deviation of reward scores could serve as a lightweight, real-time indicator for adjusting TPO depth.
>
>     By using such adaptive criteria, we can reduce computational overhead while preserving alignment quality, making TPO more efficient and practical for real-world deployment.
>
> 4. **Discussions of Limitations.**
>
>     We appreciate your suggestion to discuss the limitations and boundaries of TPO. We outline two primary limitations:
>
>     **(1) Dependence on Instruction-Following Ability:**
>     As discussed in Section 7.4, TPO assumes a base level of instruction-following capability from the policy model. For instance, TPO fails to yield improvements on weaker models such as Llama-3.1-8B-Instruct, where the model struggles to follow prompts effectively. A promising direction to mitigate this limitation is to incorporate a light form of TPO-like textual feedback into the SFT stage, thereby bootstrapping a stronger foundation for inference-time refinement.
>
>     **(2) Reward Model Limitations in Certain Domains:**
>     TPO relies heavily on the quality of the reward model (RM). In domains like coding or complex reasoning, we observe that the RM can sometimes assign higher scores to verbose yet incorrect reasoning traces, while penalizing concise but accurate responses. This misalignment introduces noise into the iterative refinement process. Future work could explore integrating more capable reward models, such as process-based RMs, to improve signal fidelity and better support structured or domain-specific tasks.
>
>     We believe these limitations outline fruitful directions for future work and practical improvement of TPO.

---

### Official Review · Reviewer_qHoH · 2025-03-12

**Overall Recommendation:** 2

**Summary:**

This paper proposes TPO, an inference-time approach that aligns LLM outputs with human preferences without updating model parameters. TPO improves model outputs by iteratively interacting with reward models, which provide rewards in textual form. The results show that the proposed approach improves LLM performance across various benchmarks compared to training-time approaches and best-of-N sampling.

**Claims And Evidence:**

Some claims in the paper need further clarification such as the efficiency.

**Essential References Not Discussed:**

Some iterative training methods are not mentioned or compared such as [1, 2].

[1] Pang, Richard Yuanzhe, et al. "Iterative reasoning preference optimization." NeurIPS 2024.

[2] Xiong, Wei, et al. "Iterative preference learning from human feedback: Bridging theory and practice for rlhf under kl-constraint." ICML 2024.

**Experimental Designs Or Analyses:**

Yes, I checked the experimental design and analyses.

**Methods And Evaluation Criteria:**

Yes, authors evaluated on a wide range of benchmarks.

**Other Comments Or Suggestions:**

N/A

**Other Strengths And Weaknesses:**

**Strengths.** This paper studies the test-time alignment problem which is an important problem for LLM test-time scaling. The results show that the proposed approach can improve LLM performance compared to training-time aligned approaches.

**Weaknesses.** My main concerns are certain claims in the paper, the lack of discussion, and some settings and results.

1. Writing: In my opinion, the authors should include textual losses and textual gradients in preliminary as this would be beneficial for readers who are not very familiar with TextGrad and directly introducing a text prompt as a loss function might be confusing (Equation 2).
2. Instead of updating the prompt $P_{\text{loss}}$​, what if an LLM is used to generate multiple $P_{\text{loss}}$​ prompts from the beginning, then generate $v_{\text{new}}$ and select the highest-scoring entry as the final response? Adding this baseline could better highlight the benefit of iterative feedback that the authors are claiming.
3. Why select only the best and worst responses in $C$? How does this compare to randomly selecting pairs of responses or choosing middle-high or middle-low responses?
4. The authors compare/discuss their approach only against static feedback methods. However, since the proposed approach is iterative, they should also compare or discuss how it differs from other iterative approaches such as [1, 2].
5. How do the authors choose the reward model? Since reward models are prone to over-optimization, how can a user select a suitable reward model and how does TPO generalize in such cases?
6. The results in Figures 3, 5, and 6 only demonstrate that TPO aligns well with the reward model whereas other methods do not directly train with the same reward model. I wonder what additional insights the authors are providing in these figures. It would be more informative if the y-axis represented downstream performance instead.
7. TPO is a test-time alignment approach, but it also introduces significant computational overhead during inference as it requires loading and performing iterative inference with both policy model and reward model. In my opinion, the comparison in Figure 5 would be fairer if the authors compared TPO with test-time methods such as BoN under the same inference time or number of FLOPs. Can the authors also report the inference time and number of FLOPs for TPO and BoN-60?

**References**

[1] Pang, Richard Yuanzhe, et al. "Iterative reasoning preference optimization." NeurIPS 2024.

[2] Xiong, Wei, et al. "Iterative preference learning from human feedback: Bridging theory and practice for rlhf under kl-constraint." ICML 2024.

**Questions For Authors:**

Please refer to the weaknesses.

**Relation To Broader Scientific Literature:**

The key contribution of this paper is the new test-time alignment approach without updating model parameters.

**Theoretical Claims:**

Yes.

---

> ### Author Rebuttal · Authors · 2025-04-01
>
> Thank you for your valuable suggestions. Below, we address your concerns concisely:
>
> 1. **Introducing TextGrad as Preliminaries.**
>
>     Thanks for your suggestion. We agree that clearly introducing TextGrad prior to discussing our methodology will enhance readability. We will include a concise overview of TextGrad in our preliminary section in the revised manuscript.
>
> 2. **TPO Variants and Ablations**
>     1. **Using Multiple** $P_{loss}$**.**
>
>         As suggested, we conducted additional experiments where we sampled five initial responses, created all possible combinations of positive and negative sample pairs, and thus generated 10 distinct  $P_{loss}$ prompts. These prompts were utilized to produce 10 new responses, and together with the initial 5, we evaluated all 15 instances using the reward model, selecting the best-performing response. We term this approach "Multiple $P_{loss}$", effectively representing breadth-first search without sequential refinement. Results (below) indicate that TPO consistently outperforms the "Multiple $P_{loss}$" variant across all tasks covering preference alignment, safety and math capability.
>
>         | Model | HH-RLHF | BeaverTails | XSTest | MATH |
>         | --- | --- | --- | --- | --- |
>         | Llama-SFT | -4.1 | -7.2 | 87.8 | 61.8 |
>         | w/  Multiple $P_{loss}$ | -1.1 | -5.2 | 89.8 | 70.8 |
>         | w/ TPO | **-0.6** | **-4.8** | **90.4** | **71.2** |
>     2. **Randomly Selecting Pairs (Middle-high and Middle-low)**
>
>         We also evaluated another variant (Mid-TPO) where pairs of mid-scoring responses (randomly chosen, ensuring the chosen response had a higher score) were used to construct $P_{loss}$. The results clearly demonstrate that TPO, utilizing the most contrasting response pairs, consistently outperforms the Mid-TPO variant.
>
>         | Model | HH-RLHF | BeaverTails | XSTest | MATH |
>         | --- | --- | --- | --- | --- |
>         | Llama-SFT | -4.1 | -7.2 | 87.8 | 61.8 |
>         | w/ Mid-TPO | -1.3 | -5.2 | 89.7 | 69.8 |
>         | w/ TPO | **-0.6** | **-4.8** | **90.4** | **71.2** |
>
>     We agree that these two variants can serve as essential ablations demonstrating the effectiveness of combining iterative refinement with parallel sampling and using contrasting samples to enhance alignment. We will include these additional experiments in our revision.
>
> 3. **Discussion of Other Iterative Approaches**
>
>     The iterative nature of TPO involves recursively revising model responses during inference *without* updating model parameters. This distinctly contrasts with the previously mentioned iterative methods [1,2], which utilize multiple rounds of iterative DPO for parameter updates. We will explicitly discuss these differences and cite the relevant works in the related work.
>
> 4. **Practice of Reward Model Selection**
>
>     Practically, reward models can be selected based on benchmarks such as Reward Bench, a widely recognized standard for evaluating reward model performance. In our study, we employed two distinct reward models: (1) FsfairX-LLaMA3-RM-v0.1 from Reward Bench, and (2) Llama-3.1-Tulu-3-8B-RM, trained from the same model family as our policy model. Empirical evidence shows that TPO effectively aligns the SFT model with both reward models, underscoring TPO’s adaptability across diverse reward frameworks.
>
> 5. **Downstream Performance (y-axis) w.r.t. TPO Steps**
>
>     In our study, we assume the reward model (e.g., FsfairX-LLaMA3-RM-v0.1) accurately approximates human preferences. Thus, higher reward scores, as presented in Figures 3, 5, and 6, indicate better alignment with human preferences. Below, we provide explicit downstream performance metrics for each TPO iteration:
>
>     | Iteration | HH-RLHF | BeaverTails | XSTest | MATH |
>     | --- | --- | --- | --- | --- |
>     | 0 | -2.8 | -6.4 | 89.78 | 69.2 |
>     | 1 | -1.3 | -5.3 | 90.00 | 69.8 |
>     | 2 | -0.5 | -4.7 | 90.44 | 71.2 |
>     | 3 | -0.1 | -4.3 | 90.44 | 71.4 |
>     | 4 | 0.2 | -4.0 | 90.22 | 71.4 |
>     | 5 | 0.4 | -3.9 | 90.66 | 71.4 |
> 6. **Reporting Inference Time and FLOPs for TPO and BoN-60**
>
>     We present a detailed comparison of computational overhead and inference time (on 4 A100 GPUs using vLLM) below. We compare the inference time on the HH-RLHF dataset based on Llama3.1-70B-SFT. The average time duration (s) for finalizing the response to each query is reported:
>
>     | Model | FLOPs (PFLOPs) | Avg. Inference Time (s/query) |
>     | --- | --- | --- |
>     | TPO-D2-N5 | 9.30 | 617/500=1.2 |
>     | BoN-60 | 16.90 | 1,384/500=2.8 |
>
>     TPO consumes substantially fewer FLOPs than BoN-60, with a lower inference latency, while achieving consistent advantages (Figure 5). Details of FLOPs computation can be found in Appendix E.
>
>
> We sincerely thank you for your valuable feedback, allowing us to enhance clarity. We hope this can address your concerns and look forward to further comments.

---

### Official Review · Reviewer_p7hk · 2025-03-13

**Overall Recommendation:** 3

**Summary:**

The paper introduces TPO, a new alignment strategy as an alternative to RLHF, that only acts at test-time, without modifying the main model's weights. In particular, TPO works by learning a proxy reward model, and iterating between generating N completions from the models, ranking them, and formatting the best and worse rewarded completion into a 'feedback prompt'. The authors show results, outperforming traditional a training-time aligned Llama and Mistral models after applying their method on an SFT-only counterpart.

**Claims And Evidence:**

There is a clear nice effort in providing evidence that TPO could be used in place of RLHF. The main Llama and Mistral models considered are very relevant, for both researchers and practitioners, and the benchmarks chosen are also competitive and cover a wide range of domains.

**Essential References Not Discussed:**

While there are some issues in how references are discussed, I believe a good part of the most relevant literature is mentioned.

**Experimental Designs Or Analyses:**

The main claims in the paper are mostly supported by the experiments, as mentioned above. However, I believe there are a few clear areas of improvement:

Given the extra computation allowed in TPO, I think that simply using Llama 70B instruct and Mistral instruct sampling the same number of responses like TPO and selecting the best one according to the employed PRM should have also be reported as results for all the tasks along with the other main results in Tables 1 and 2.

A comparison against other prior test-time alignment methods (e.g., ICDPO) is currently missing and would have been a relevant inclusion.

**Methods And Evaluation Criteria:**

Overall, the methodology proposed is simple and intuitive.

A key practical downside of the methodology is the increase in computational budget. At test time, the model needs to generate a lot of different responses which cannot be obtained in parallel, as the proposed approach employs multiple stages of scoring, prompt refinement, etc. However, I appreciated that the authors do touch on this limitation at the end of the paper.

**Other Comments Or Suggestions:**

Since at the end of the proposed process the model's response with the highest reward predicted by the PRM is employed, I think providing exact statistics of the amount of responses belonging to initial sampling stages would have been helpful.

---

Post rebuttal: Given the additional response and clarifications, I will vote for acceptance. However, the latest ICDPO results are very much not apples-to-apples. Thus, I would like to ask the authors to make sure these are well-contextualized within to text to avoid misleading 'outperforming' claims in future revisions.

**Other Strengths And Weaknesses:**

While there are several unnecessary, potentially confusing and unnecessary equations (see Theoretical claims), the paper is still mostly clearly written.

**Questions For Authors:**

No additional questions.

**Relation To Broader Scientific Literature:**

I think the closest prior work is ICDPO [1], which also proposed a fully-online alignment strategy by providing the model in-context examples of aligned responses that can be obtained from external sources. I think there should be a greater emphasis on explaining the differences with this method for an unfamiliar audience. Lines 84-86 do not provide any details, only briefly mentioning the work by name.

[1] Song, Feifan, et al. "Icdpo: Effectively borrowing alignment capability of others via in-context direct preference optimization." arXiv preprint arXiv:2402.09320 (2024).

**Theoretical Claims:**

Several equations and the overall formal notation in the paper do not seem to make much sense and should definitely be revised and fixed. For instance, the 'main objective' said to be optimized by alignment methods in line 141 is a constant value with respect to the dataset and scoring function, with no dependency on the model's parameters. Furthermore, I would really suggest against excessive abuse of gradient/differential notations when the paper does not involve any actual new math (e.g., Equation 3).

---

> ### Author Rebuttal · Authors · 2025-04-01
>
> Thank you for your insightful comments. Below, we address your concerns concisely:
>
> 1. **Revision of Overall Formal Notation**
>
>     We acknowledge the confusion caused by the notation in Equation (3) and we will carefully revise these equations. Additionally, we will reduce the use of gradient/differential notation to avoid confusion, reserving it strictly for illustrative analogy.
>
> 2. **Comparison with BoN on more Tasks**
>
>     Following previous work, we compared TPO with BoN using GPT4 win-rate in **Section 7.2.**
>
>     As suggested, we conducted evaluations across a broader set of benchmarks, comparing two methods with the same number of responses. As shown below, TPO consistently outperforms BoN across all models and tasks, reinforcing the findings presented in Section 7.2.
>
>     **Llama3.1-70B-SFT:**
>
>     | Model | ArenaHard | HH-RLHF | BeaverTails | XSTest |
>     | --- | --- | --- | --- | --- |
>     | BoN | 63.6 | -2.1 | -6.2 | 89.8 |
>     | TPO | **69.7** | **0.6** | **-4.8** | **90.4** |
>
>     **Llama3.1-70B-Instruct:**
>
>     | Model | ArenaHard | HH-RLHF | BeaverTails | XSTest |
>     | --- | --- | --- | --- | --- |
>     | BoN | 68.7 | 1.0 | -4.5 | 88.7 |
>     | TPO | **69.5** | **1.3** | **-3.6** | **89.6** |
>
>     **Mistral-Small:**
>
>     | Model | ArenaHard | HH-RLHF | BeaverTails | XSTest |
>     | --- | --- | --- | --- | --- |
>     | BoN | 68.3 | 0.9 | -3.7 | 90.6 |
>     | TPO | **72.2** | **1.1** | **-3.4** | **90.7** |
> 3. **Comparison with ICDPO**
>
>     We will expand our discussion of ICDPO in the related work section. Additionally, we implemented ICDPO using SBERT-based demonstration retrieval as described in their paper and GitHub repository. We aligned the number of sampled responses with TPO and evaluated both methods on Llama3.1-70B-SFT. The results are summarized below:
>
>     | Model | ArenaHard | HH-RLHF | XSTest |
>     | --- | --- | --- | --- |
>     | TPO | 69.7 | 0.6 | 90.4 |
>     | ICDPO | 6.6 | -5.3 | 70.4 |
>
>     TPO consistently outperforms ICDPO across tasks including instruction following, preference alignment and safety. We will include this comparison in our revision.
>
> 4. **Statistics of the Amount of Responses Belonging to Initial Sampling Stages**
>
>     We appreciate your valuable suggestions, which are essential to strengthen our claims for combining parallel search with sequential revision.
>
>     Below, we present the distribution of best-scored responses across each TPO step, with "TPO step 0" indicating the initial stage (equivalent to a Best-of-N candidate search).
>
>     Overall, best-scored candidates increasingly emerge from later TPO steps (those involving more revisions), especially evident in the unaligned SFT model. Notably, the initial stage typically yields the lowest proportion of optimal responses, which progressively increases through subsequent TPO iterations.
>
>     In contrast, aligned models exhibit a flatter distribution, suggesting these models already generate sufficiently high-quality responses, thus requiring fewer refinement iterations.
>
>     **Llama3.1-70B-SFT:**
>
>     | Testset | TPO step 0 | TPO step 1 | TPO step 2 | TPO step 3 | TPO step 4 | TPO step 5 |
>     | --- | --- | --- | --- | --- | --- | --- |
>     | AlpacaEval | 2.0% | 7.2% | 9.44% | 17.7% | 26.1% | 37.6% |
>     | ArenaHard | 2.6% | 11.2% | 12.0% | 18.8% | 21.4% | 34.0% |
>     | HH-RLHF | 0.4% | 5.0% | 7.8% | 13.6% | 26.4% | 46.8% |
>     | BeaverTails | 0.0% | 3.29% | 7.43% | 13.86% | 26.29% | 49.14% |
>     | XSTest | 0.2% | 4.2% | 8.2% | 16.7% | 27.1% | 43.6% |
>     | MATH | 4.2% | 5.6% | 11.4% | 18.2% | 24.0% | 36.6% |
>     | Average | 1.6% | 6.8% | 9.7% | 16.8% | 25.6% | **40.0%** |
>
>     **Llama3.1-70B-Instruct:**
>
>     | Testset | TPO step 0 | TPO step 1 | TPO step 2 | TPO step 3 | TPO step 4 | TPO step 5 |
>     | --- | --- | --- | --- | --- | --- | --- |
>     | AlpacaEval | 12.2% | 11.8% | 13.8% | 15.8% | 20.7% | 25.7% |
>     | ArenaHard | 20.6% | 10.4% | 13.0% | 14.0% | 16.0% | 26.0% |
>     | HH-RLHF | 8.4% | 12.0% | 15.4% | 14.8% | 20.8% | 28.6% |
>     | BeaverTails | 1.7% | 10.0% | 11.3% | 16.4% | 23.86% | 36.71% |
>     | XSTest | 4.7% | 12.4% | 10.4% | 12.0% | 23.3% | 37.1% |
>     | MATH | 10.0% | 6.0% | 9.0% | 18.8% | 23.2% | 33.0% |
>     | Average | 9.6% | 10.4% | 12.2% | 15.3% | 21.6% | **31.5%** |
>
>     **Mistral-Small:**
>
>     | Testset | TPO step 0 | TPO step 1 | TPO step 2 | TPO step 3 | TPO step 4 | TPO step 5 |
>     | --- | --- | --- | --- | --- | --- | --- |
>     | AlpacaEval | 8.8% | 10.2% | 10.8% | 13.8% | 26.34% | 30.1% |
>     | ArenaHard | 10.4% | 9.4% | 10.6% | 16.8% | 21.2% | 31.6% |
>     | HH-RLHF | 11.2% | 6.6% | 11.6% | 17.0% | 23.6% | 30.0% |
>     | BeaverTails | 6.7% | 8.6% | 13.0% | 18.0% | 21.4% | 32.3% |
>     | XSTest | 10.0% | 9.8% | 9.6% | 15.6% | 22.4% | 32.7% |
>     | MATH | 7.8% | 9.2% | 9.8% | 13.4% | 24.2% | 35.6% |
>     | Average | 9.2% | 9.0% | 10.9% | 15.8% | 23.2% | **32.1%** |

---

> > ### Comment · Reviewer_p7hk · 2025-04-05
> >
> > I thank the authors for their rebuttal. However, I am very much not convinced by the ICDPO results. Performance is crashing far below the original model. Can the authors provide an explanation for why this method seems even harmful, in clear contrast to what would be intuitively expected, and the results in the ICDPO paper?

---

> > > ### Author Response · Authors · 2025-04-05
> > >
> > > Thank you for taking the time to review our rebuttal and provide timely feedback. We understand your concerns regarding the ICDPO results, and we agree that the observed performance drop is surprising, especially given the positive outcomes reported in the original ICDPO paper. Below, we outline our efforts and analysis to clarify this discrepancy.
> > >
> > > We carefully implemented ICDPO in our experimental setup, exploring multiple variants mentioned in the original paper, including both random retrieval and SBERT-based demonstration retrieval. The results we reported reflect the best-performing configuration among these variants.
> > >
> > > We believe there are two main reasons for the discrepancy:
> > >
> > > **1. Different LLM Backbones: Base Models v.s. Instruction-Tuned Models.**
> > >
> > > In the original ICDPO paper, test-time alignment is applied to **base models** such as Llama-base and Mistral-base. In contrast, our evaluation uses **instruction-tuned models** (e.g., Llama3.1-70B-SFT) to align with the results of TPO. ICDPO relies on the model’s internal confidence to distinguish better from worse responses, assuming it is well-calibrated, i.e., able to assign higher probabilities to better responses with appropriate certainty, which is an ability typically found in base models [1–3]. However, several studies [1–3] have shown that instruction tuning significantly degrades calibration, making it more difficult for SFT models to reliably distinguish between good and bad responses based on their own confidence. As a result, selecting responses based on the SFT model’s own confidence can lead to suboptimal or even harmful choices. This also helps explain why Best-of-N sampling is less effective in base models (as shown in the ICDPO paper) but more effective post-SFT, when external reward models like the one used in BoN or TPO become essential.
> > >
> > > **2. In-Context Learning Benefits Diminish After SFT.**
> > >
> > > A core strength of ICDPO lies in in-context learning. However, prior work [4,5] suggests that SFT already instills many of the benefits typically gained through ICL. Thus, applying ICDPO to SFT models yields diminished returns compared to its original use case on base models.
> > >
> > > In summary, ICDPO and TPO are **complementary** test-time alignment approaches that operate at different stages of the LLM lifecycle. ICDPO is more suitable for aligning base models, while TPO is designed for models that have already undergone instruction tuning. Indeed, as discussed in Section 7.4, TPO itself requires a minimal level of instruction-following ability, and thus cannot operate effectively on base models where ICDPO demonstrates strong performance.
> > >
> > > We appreciate your feedback and the opportunity to clarify our findings. We will include this analysis and discussion in the revised version of our paper.
> > >
> > > **References:**
> > >
> > > [1] *On the Calibration of Large Language Models and Alignment*
> > >
> > > [2] *Investigating Uncertainty Calibration of Aligned Language Models under the Multiple-Choice Setting*
> > >
> > > [3] *Calibrated Language Model Fine-Tuning for In- and Out-of-Distribution Data*
> > >
> > > [4] *Exploring the Relationship between In-Context Learning and Instruction Tuning*
> > >
> > > [5] *Preserving In-Context Learning Ability in Large Language Model Fine-Tuning*

---

### Official Review · Reviewer_qXZY · 2025-03-14

**Overall Recommendation:** 2

**Summary:**

This paper studies test-time preference optimization by finding responses that maximize the reward model values with verbal reinforcement learning.

**Claims And Evidence:**

Yes, the claims are well-supported.

**Essential References Not Discussed:**

The related works are well covered.

**Experimental Designs Or Analyses:**

The experiments show the effectiveness of the proposed method and compare important baselines including BoN. However, several critical baselines are still missing, which makes the results less convincing. Firstly, since the method adopts test-time verbal RL, I would like to see the comparison with directly applying it during training time, such as [1]. This can be implemented by e.g., generating summaries of experiences by trial and error on training alignment data and appending them at test time. This would significantly reduce the required compute. So it would be great if the authors can include winning rate versus compute curves comparing with [1]. I would also like to see comparisons with training-based alignment methods with parameter tuning, such as DPO.

[1] Shinn et al., "Reflexion: Language Agents with Verbal Reinforcement Learning."

**Methods And Evaluation Criteria:**

Yes, the instruction-following benchmarks are commonly used in alignment experiments.

**Other Comments Or Suggestions:**

I didn't find obvious typos.

**Other Strengths And Weaknesses:**

The paper is very clearly written. But some experiment baselines are missing.

**Questions For Authors:**

Could the authors explain why they select 100 instances from AlpacaEval 2, Arena-Hard, and HH-RLHF instead of the full benchmark so that future experiments can reproduce? I will consider raising my score if the suggested experiments are added.

**Relation To Broader Scientific Literature:**

It is related to alignment, instruction following, and test-time scaling.

**Theoretical Claims:**

N/A. No theory in this paper.

---

> ### Author Rebuttal · Authors · 2025-04-01
>
> Thank you for your insightful comments and valuable suggestions. Below, we address your concerns concisely:
>
> 1. **Comparison with Test-time Verbal RL**
>
>     We appreciate your suggestions providing an opportunity to compare with test-time verbal RL. As recommended, we implemented Reflexion, which integrates summary-based verbal feedback, and compared it to TPO (D2-N5) in terms of both performance and computational cost. The table below presents results across multiple iterations:
>
>     | Iteration | FLOPs (TPO) | PFLOPs (reflexion) | PFLOPs ratio (TPO against Reflexion) | # LLM Calls (TPO/Reflexion) | ArenaHard: GPT4 Win-rate (TPO against Reflexion) | AlpacaEval: GPT4 Win-rate (TPO against Reflexion) | HH-RLHF scores (TPO/Reflexion) |
>     | --- | --- | --- | --- | --- | --- | --- | --- |
>     | 0 | 1408.5 | 5070.6 | 0.2778 | 5/9 | 91.50 | 94.16 | -3.0/-8.4 |
>     | 1 | 5352.3 | 10704.6 | 0.5000 | 12/19 | 92.40 | 94.74 | -1.5/-7.9 |
>     | 2 | 9296.1 | 16338.6 | 0.5690 | 19/29 | 92.13 | 94.65 | -0.7/-7.5 |
>
>     As can be seen, TPO consistently and substantially outperforms Reflexion, with a lower computational overhead, highlighting the benefits of integrating parallel search with sequential refinement. The details of FOLPs calculation are presented below:
>
>     In the Reflexion framework, one Reflexion iteration consists of multiple cycles (3 on average) of “Thought–Action–Observation”  along with a final reflection step, where each action represents an LLM call. Given an iteration number of D, the FLOPs can be approximately calculated as: (3\*3+1)\*D\*F + 3\*3\*F, where F is the FLOPs of an LLM call and the last iteration requires no reflection. To support and accommodate complex reasoning and integration of auxiliary contextual information, we set the maximum generation length as 4,096 tokens.  According to estimates from calflops, generating 4,096 tokens requires approximately 563.4 TFLOPs. For FLOPs computation of TPO, please refer to Appendix E.
>
> 2. **Comparison with Training-time Alignment Methods like DPO**
>
>     In **Section 6**, we indeed compared our method with two types of training-time alignment methods: (1) Llama3.1-70B-instruct and (2) Llama3.1-70B-DPO. We attach the results from our paper:
>
>     | Model | ArenaHard | AlpacaEval | HH-RLHF | BeaverTails | XSTest | Math |
>     | --- | --- | --- | --- | --- | --- | --- |
>     | Llama-3.1-70B-DPO | 32.3/23.1 | 50.4 | -2.8 | -6.7 | 89.8 | 63.4 |
>     | Llama-3.1-70B-TPO | **33.0/40.5** | **69.7** |  **-0.6** | **-4.8** | **90.4** | **71.2** |
>
>     Llama3.1-70B-DPO was trained using DPO on Llama3.1-70B-SFT, the same model for test-time alignment, i.e., TPO. Detailed experimental settings can be found in **Section 5** and **Appendix C**. This model, i.e., Llama3.1-70B-DPO, was designed to serve as a fair testbed to compare preference alignment during training-time and test-time. Intuitively, DPO stores preference information in model weights, while TPO encodes alignment in the context (KV cache), allowing for flexible, test-time adaptation without retraining. We provide an in-depth comparison of TPO and DPO, including their computational overheads, in **Section 7.3**.
>
> 3. **The Reasons of Selecting 100 Instances in Section 7.2**
>
>     The use of 100 instances to compute win rates against BoN follows established practice in prior work [1, 2].
>
>     We present the evaluation of several complete testsets (covering instruction following, preference alignment, and safety) below, comparing TPO with BoN with the same number of responses:
>
>     **Llama3.1-70B-SFT:**
>
>     | Model | ArenaHard | HH-RLHF | BeaverTails | XSTest |
>     | --- | --- | --- | --- | --- |
>     | BoN | 63.6 | -2.1 | -6.2 | 89.8 |
>     | TPO | **69.7** | **0.6** | **-4.8** | **90.4** |
>
>     **Llama3.1-70B-Instruct:**
>
>     | Model | ArenaHard | HH-RLHF | BeaverTails | XSTest |
>     | --- | --- | --- | --- | --- |
>     | BoN | 68.7 | 1.0 | -4.5 | 88.7 |
>     | TPO | **69.5** | **1.3** | **-3.6** | **89.6** |
>
>     **Mistral-Small**
>
>     | Model | ArenaHard | HH-RLHF | BeaverTails | XSTest |
>     | --- | --- | --- | --- | --- |
>     | BoN | 68.3 | 0.9 | -3.7 | 90.6 |
>     | TPO | **72.2** | **1.1** | **-3.4** | **90.7** |
>
>     [1] Accelerating Best-of-N via Speculative Rejection
>
>     [2] TreeBoN: Enhancing Inference-Time Alignment with Speculative Tree-Search and Best-of-N Sampling

---

### Decision · Program_Chairs · 2025-05-01

**Decision:**

Accept (poster)

**Comment:**

This submission proposed a test-time preference alignment method that aligns LLM outputs with human preferences during inference. The key idea is to make use of iterative textual feedback: the proposed method translates reward signals into textual critiques and uses them as textual rewards to iteratively refine the LLM response. Although the reviewers gave relatively low scores, I did not find major concerns (perhaps they had more concerns not clearly expressed in their reviews) that were still unaddressed after the deadline of the author-reviewer discussion.

The submission showed that for some popular open-source LLMs after SFT (SFT-only, no RLHF or DPO yet), test-time preference alignment can be even better than training-time preference alignment. Based on my personal experience, when we can touch the model weights, manipulating the model weights is more effective than manipulating the prompt for the problems I have studied so far. If the results in this paper are all correct (and I assumed so), it might become an important paper about preference alignment in the near future. Consequently, it is a good idea to first accept it for publication and then let the LLM community check the claims.